# Intrathecal versus Peripheral Inflammatory Protein Profile in MS Patients at Diagnosis: A Comprehensive Investigation on Serum and CSF

**DOI:** 10.3390/ijms24043768

**Published:** 2023-02-13

**Authors:** Francesco Pezzini, Annalisa Pisani, Valentina Mazziotti, Damiano Marastoni, Agnese Tamanti, Edilio Borroni, Stefano Magon, Bastian Zinnhardt, Roberta Magliozzi, Massimiliano Calabrese

**Affiliations:** 1Department of Neurosciences, Biomedicine and Movement Sciences, University of Verona, 37134 Verona, Italy; 2Department of Surgery, Dentistry, Paediatrics and Gynaecology, University of Verona, 37134 Verona, Italy; 3Roche Pharma Research & Early Development (pRED), Biomarkers & Translational Technologies (BTT), F. Hoffmann-La Roche Ltd., CH-4070 Basel, Switzerland

**Keywords:** multiple sclerosis, inflammation, CSF and serum, biomarkers

## Abstract

Intrathecal inflammation plays a key role in the pathogenesis of multiple sclerosis (MS). To better elucidate its relationship with peripheral inflammation, we investigated the correlation between cerebrospinal fluid (CSF) and serum levels of 61 inflammatory proteins. Paired CSF and serum samples were collected from 143 treatment-naïve MS patients at diagnosis. A customized panel of 61 inflammatory molecules was analyzed by a multiplex immunoassay. Correlations between serum and CSF expression levels for each molecule were performed by Spearman’s method. The expression of sixteen CSF proteins correlated with their serum expression (*p*-value < 0.001): only five molecules (CXCL9, sTNFR2, IFNα2, Pentraxin-3, and TSLP) showed a Rho value >0.40, suggesting moderate CSF/serum correlation. No correlation between inflammatory serum patterns and Q_alb_ was observed. Correlation analysis of serum expression levels of these sixteen proteins with clinical and MRI parameters pinpointed a subset of five molecules (CXCL9, sTNFR2, IFNα2, IFNβ, and TSLP) negatively correlating with spinal cord lesion volume. However, following FDR correction, only the correlation of CXCL9 remained significant. Our data support the hypothesis that the intrathecal inflammation in MS only partially associates with the peripheral one, except for the expression of some immunomodulators that might have a key role in the initial MS immune response.

## 1. Introduction

The early and exact diagnosis of neurological diseases is challenging, particularly for multiple sclerosis (MS), due to its pathological and clinical heterogeneity among patients. The need for advanced and specific diagnostic procedures allowed the proposal of several new MS biomarkers, including fluid, imaging, genetic, and immunogenetic biomarkers [1,2]. In particular, different body fluids, such as the serum and cerebrospinal fluid (CSF), have been proposed to provide novel diagnostic and prognostic value in MS and could be potentially used in clinical practice to support subtyping and staging of MS. 

Several experimental and clinical studies demonstrated that the assessment of CSF profiling could specifically reflect intrathecal inflammatory events characterizing MS since early disease phases [3,4,5].

However, as obtaining CSF from MS patients remains difficult and not always adopted in clinical practice, it is important to compare the results obtained by CSF studies with a paired serum analysis [6]. In addition, the convenience of periodically repeating the serum analysis to monitor biomarkers dynamics and its correlation with disability accumulation and treatment response further support this need.

Recent optimization of advanced techniques for quantitative protein measurements demonstrated that CSF and serum neurofilament light chain (sNfL) concentrations in MS patients are highly correlated, thus sNfL measurements are rapidly gaining traction as a potential biomarker for MS in numerous studies and clinical trials [7,8].

Our recent studies showed that a specific CSF molecular pattern is associated with an increased intrathecal (meninges and CSF) inflammatory milieu, with cortical gray matter lesion load and more severe progression either at diagnosis or at the time of death [9,10]. In addition, we recently proposed that the analysis of CSF protein patterns differentially reflects the degree of cognitive impairment [11], primary progressive course [12], as well as specific deep gray matter atrophy [13].

To clarify whether these specific CSF patterns, mainly reflecting intrathecal inflammatory changes [9], correlate with the serum biomarkers pattern, we performed a comprehensive multiplex immunoassay in paired CSF and serum samples obtained at the time of diagnosis in association with a detailed clinical and MRI assessment. This study may not only verify the hypothesis of the potential correlation between intrathecal and peripheric MS inflammation but also suggest specific novel serum biomarkers for disease diagnosis and monitoring.

## 2. Results

### 2.1. Correlation Analysis between Liquoral and Serum Expression Levels of 61 Inflammatory Mediators

We identified 16 molecules showing a significant correlation between intrathecal (liquor) and peripheral (serum) expression levels in MS patients (Figure 1A). The majority showed a positive correlation; only CXCL16 showed a negative correlation (Rho = −0.23, *p* < 0.0001). Five molecules (CXCL9, sTNFR2, IFNα2, Pentraxin-3, and TSLP) exhibited a Rho value higher than 0.40, indicating a moderate correlation between serum and liquor expression, whereas the remaining molecules showed a Rho < 0.40 (Figure 1B). No significant correlation with BBB damage quantified by the Q_alb_ value was observed.

### 2.2. Correlations between Serum-CSF Profiles and Clinical-MRI Parameters of MS Patients at the Time of Diagnosis

The serum expression levels of CXCL9, sTNFR2, IFNα2, IFNβ, and TSLP negatively correlated with spinal cord lesions (Rho = −0.35, *p*-value < 0.001; Rho = −0.20, *p*-value = 0.03; Rho = −0.25, *p*-value < 0.01; Rho = −0.28, *p*-value < 0.01; and Rho = −0.23, *p*-value < 0.05, respectively); IFNα2 and TSLP also negatively correlated with Gad+ lesions (*p*-value < 0.05). In the CSF, only the expression of APRIL positively correlated with spinal cord lesions (Rho = 0.20, *p*-value < 0.05). The CSF expression levels of the other ten molecules (CCL24, CCL22, CXCL9, CCL25, sTNFR2, INFα2, IFNβ, MMP1, Pentraxin-3, and TSLP) showed positive correlations with the number and the volume of cortical lesions, and T2 white matter lesions (T2WML). However, after FDR correction, most of the reported correlations became not significant. Only two molecules, serum CXCL9 and CSF IFNβ, showed a significant correlation with spinal cord lesions (Rho = −0.35 *p* < 0.001) and with T2MLV (Rho = 0.33 *p* < 0.001), respectively (black squares in Figure 2). 

### 2.3. Serum Expression Level of the 16 Molecules in MS Patients Compared to OND and HS

We then compared the serum levels of the identified 16 molecules among patients with MS and other neurological disorders (OND) and healthy subjects (HS). Compared to both OND and HS groups, in MS patients the expression of three proteins (TSLP, IFNβ, and CXCL16) was found to increase, whereas only the sIL-6Rα expression was decreased (Table 1). Furthermore, compared to OND only, we observed lower levels of Pentraxin-3, CCL25, gp130/sIL-6Rβ, and sCD30 in MS patients. Finally, the comparison between MS and HS pinpointed four up-regulated proteins (sTNF-R2, CCL24, MMP-1, and CCL22) and one down-regulated protein (IFNα2) in MS patients.

### 2.4. Comparison of Serum Expression Levels between RR, PP, and CIS MS Subgroups

Following stratification of patients according to the clinical phenotype at the time of diagnosis (clinically isolated syndrome: CIS; primary progressive MS: PPMS; relapsing-remitting MS: RRMS) we found that the serum expression levels of CXCL9, IFNα2, TSLP, CCL24, and CCL22 were increased in CIS patients compared to RRMS patients (Figure 3). Only CCL25 was found to be increased in patients with RRMS compared to those with a CIS. No molecules in these panels were differentially expressed between the RR and PP groups.

## 3. Discussion

Several clinical and experimental studies have extensively indicated that intrathecal inflammation, possibly compartmentalized in central nervous system (CNS) niches, such as meninges, CSF, choroid plexus, and perivenular spaces, may represent one of the main drivers of progression in MS pathology [3,4]. This suggests the importance of monitoring inflammatory biomarker expression directly in the CSF, which may predict several disease features, including cognitive impairment [5,9,10,11,14]. However, the potential correlation between the intrathecal inflammation specifically reflected by CSF inflammatory profile and the periphery one linked to the blood profile remains unclear.

In this cross-sectional study, the evaluation of 61 inflammatory proteins in paired CSF and serum in a large cohort of naïve-treatment MS patients at the time of diagnosis revealed a poor correlation between the inflammatory proteomic profile of CSF and the corresponding serum. Only sixteen molecules showed a significant correlation between the two compartments, and only five of these molecules (CXCL9, sTNFR2, IFNα2, Pentraxin-3, and TSLP) showed mild correlations (Rho > 0.40), whereas for the remaining eleven molecules the correlations were not substantial (Rho < 0.40). 

These data provide further evidence that the CSF inflammatory profile may reflect different disease mechanisms that are not clearly reflected in the serum, probably mainly related to the intrathecal expression of immunomodulators expressed and produced either by cells infiltrating the CNS or resident-activated glial cells.

If confirmed, such a hypothesis strongly suggests the relevant role of the assessment of the CSF inflammatory profile at the disease’s onset, reflecting ongoing intrathecal inflammation. CSF molecular profiling may therefore improve the accuracy of diagnosis and help in treatment tailoring [5].

We identified CSF-serum correlation for the expression of 16 molecules that, however, were scarcely correlated with the clinical and MRI parameters evaluated at the diagnosis. Furthermore, none of the serum molecules, such as CXCL9, sTNFR2, IFNα2, IFNβ, and TSLP, which were associated with spinal cord lesions and Gad+ lesions, were correlated with the same parameters considering their CSF expression. 

It is challenging to explain the potential biological/pathological reason for the negative correlation between serum expression of some inflammatory proteins and the number of spinal cord lesions or Gad+ lesions. Our finding highlights the possibility of early identification of MS patients with a high risk of disease activity and disability accumulation by measuring the serum levels of this inflammatory pattern, together with the use of appropriate imaging tools. In particular, at the time of diagnosis, MS patients are probably characterized by a specific immune-pathogenic profile that may contribute to the prevalence of correspondent disease phenotype. It would be, therefore, interesting to either validate our findings in larger and independent MS cohorts or further analyze the clinical-imaging follow-up of the examined MS patients.

Among the five identified molecules, TSLP and INFβ were the only two molecules whose serum expression was increased in MS patients compared to OND and HS. Moreover, the expression of TSLP in serum was able to discriminate between RR and CIS patients. It may, therefore, be suggested that in the initial stage of the disease, these two molecules in the serum may play a key role in regulating the potential resolution and/or activation of the inflammatory processes, including, in particular, the T cell-mediated immune response. A longitudinal follow-up study of the predictive role of these molecules, as well as the validation in an independent MS group, should clarify whether their highest levels at the time of diagnosis may be potentially linked to better disease outcomes.

Even if INFβ and IFNα2 showed an opposite expression in the serum of MS patients compared to healthy controls (i.e., IFNβ increased, IFNα2 decreased), their serum levels would correlate with the number of spinal cord lesions. On the contrary, the same biomarkers in the CSF were associated with T2WMLV. It would be interesting to understand better why these type I IFNs may reflect different imaging correlates according to the intrathecal (CSF) or peripheral (blood) expression. However, these data strongly support that these molecules play a key role at the beginning of MS inflammatory response. Several studies have previously proposed the importance of type I IFNs in multifunctional immunomodulatory, antiviral, antiproliferative, and anti-inflammatory functions in MS and other chronic inflammatory diseases such as chronic viral hepatitis, experimental colitis, experimental allergic encephalomyelitis, experimental arthritis, and neonatal inflammation [15,16].

Among the identified molecules present both in CSF and serum samples of examined MS patients, four (IFNα2, APRIL, Pentraxin-3, and CCL25) were previously described as part of a CSF inflammatory pattern predictive of disease progression and severe cortical damage in MS patients after four years of clinical and imaging follow-up from the diagnosis [10]. Therefore, this result may suggest that the assessment at the time of diagnosis of serum levels (not only CSF ones) of IFNα2 and CCL25 (previously found to be associated with new WM lesions) and APRIL (found to be associated with increased annual cortical thinning) may help to predict the clinical outcome.

However, most of these identified correlations were not strong and became not significant after the FDR correlation. Further validation and functional studies are therefore mandatory to better understand this issue. However, following FDR correction, only the correlation of CXCL9 remained significant, suggesting a key role of this molecule in the spinal cord tissue damage. It is known that CXCL9, particularly overexpressed by stromal cells under inflammatory conditions, is involved, together with other lymphoid chemokines such as CXCL10, CXCL11, CCL2, CCL3, CCL5, and CCL5, in the initial local recruitment of peripherally primed lymphocytes into the target tissues [17]. It would be relevant in the future to further analyze the potential cellular and molecular mechanisms of spinal cord tissue damage specifically mediated by CXCL9.

It is well known that the degree of BBB damage may influence the correlation between intrathecal and peripheral inflammation. Still, it remains to be better elucidated how this may contribute to the direct diffusion/interaction of inflammatory proteins of different molecular weights and solubility features and inflammatory cell sources of intrathecal inflammation [18,19]. In our cohort, we did not observe any correlation between inflammatory serum patterns and Q_alb_; the majority of MS patients in our cohort showed, in fact, a Q_alb_ below the threshold of 7 (Table 2), indicating that there was no BBB damage at time of serum and CSF collection. These findings support the importance of carrying out the lumbar puncture far from the clinical relapse event to avoid interference from the BBB damage.

We are aware that this study may be considered explorative and has some limitations: (i) serum analysis may be influenced by several other factors even from other peripheral tissues, which may confound the analysis [20]; (ii) the number of healthy and of inflammatory controls should be increased; (iii) our MS cohort is unbalanced, since RRMS cases are the most prominent group compared to PP and CIS, suggesting the need to investigate a larger and independent MS cohort in order to better analyze putative differences in the expression of inflammatory molecules among the three MS classes.

In conclusion, this study indicates that the CSF protein profile better reflects intrathecal inflammation and only partially may be associated with the peripheral one. However, serum inflammatory profile seems to correlate with specific MS features such as spinal cord damage and could therefore be helpful for disease diagnosis and prognosis, but also to monitoring therapeutic efficacy. Thus, it deserves further investigation that will help to identify useful blood biomarkers for diagnosis and monitoring.

## 4. Materials and Methods

### 4.1. MS Patient Cohort and Clinical Evaluation

Paired CSF and serum samples were collected from 143 naïve patients at first neurological and radiological assessment, at the time of the clinical onset, (and at least two months after the last relapse and the cortisone treatment; Table 2) before starting any treatment. The inclusion criteria to be enrolled in the study were: (i) to come to our medical center consecutively from 2017 to 2020; (ii) to receive a confirmed diagnosis of MS. The exclusion criteria were: (i) any disease-modifying treatment (DMT) or any other therapy targeting the immune system, ongoing at the time of lumbar puncture; (ii) the concurrence of any other immunological or hematological diseases or any chronic infectious diseases; (iii) cortisone treatment within two months before the lumbar puncture; (iv) pregnancy. MS phenotype (CIS, PPMS, and RRMS) was defined by the neurologists of the group (blinded with respect to the CSF/serum analyses) at the time of the first diagnostic assessment including lumbar puncture, which is routinely performed during the initial work-up [21]. For comparative analysis, the serums of 30 patients affected with other neurological disorders at the time of diagnosis, before any pharmacological treatment, were also examined: these patients include 16 individuals with other inflammatory diseases (age at onset: 46.5 ± 11.04) and 14 with non-inflammatory diseases (age at onset: 45.5 ± 10.04). The other neurological inflammatory diseases included, among others, NMOSD, anti-MOG encephalomyelitis, recurrent inflammatory optic neuritis, and recurrent inflammatory myelitis; the other neurological not-inflammatory diseases included vascular encephalopathy, peripheral neuropathy, ALS, olivopontocerebellar atrophy, and essential tremor. In addition, serum from 18 age-matched healthy subjects (HS) was also utilized. Brain–blood barrier (BBB) alteration in MS patients was quantified as Q_alb_ (CSF_albumin_/Serum_albumin_).

### 4.2. MRI 

Brain and spinal cord MRI was carried out for each MS patient using a 3T Philips Achieva scanner (Philips Healthcare, Best, The Netherlands). The protocol was: 3D T1-weighted (with a repetition time TR = 8.4 ms, echo time TE = 3.7 ms, voxel size = 1 × 1 × 1 mm^3^) acquired before and after intravenous gadolinium injection; 3D fluid attenuated inversion recovery (FLAIR with TR/TE = 8000/288 ms and inversion time TI = 2360 ms, voxel size = 1 × 1 × 1 mm^3^); 3D double inversion recovery (DIR with TR/TE = 5500/300 ms, TI1/TI2 = 450 ms/2550 m, voxel size = 1 × 1 × 1 mm^3^); sagittal short-TI inversion recovery (STIR with TR/TE = 4000/50 ms, 0.4 × 0.4 × 2.5 mm); sagittal T2-weighted (with TR/TE = 3500/120 ms, voxel size = 0.9 × 0.9 × 0.9 mm^3^); and axial T2*weighted (TR/TE/ΔTE = 600/7.2/6.5 ms, three echoes, 0.5 × 0.5 × 5 mm) images acquired at the locations of the lesions identified on the sagittal images.

White and gray matter lesions were identified using FLAIR and DIR images, respectively, following the recommendations for MS lesions assessment [22,23]. WM lesions were segmented to obtain the obtaining T2 hyperintense WM lesion volume (T2WMLV) using the lesion segmentation tool (LST) algorithms after manual revision by an observer with significant experience in MS. The cortical lesion volume was assessed using a semiautomatic thresholding technique based on a Fuzzy C-mean algorithm included in the Medical Images Processing, Analysis and Visualization toolbox (http://mipav.cit.nih.gov).

### 4.3. Serum and CSF Immunoassay Protein Analysis

Paired CSF and serum samples were collected at least 2 months after the onset of clinical symptoms(s) and immediately processed according to consensus guidelines for CSF and blood biobanking [24]. After centrifugation of CSF, the supernatant and the cell pellet were stored separately at −80 °C. A customized panel of 73 inflammatory molecules was analyzed by bead-based multiplex immunoassays (40-Plex and 37-Plex, Bio-Plex X200 System, BioRad, Hercules, CA, USA) in both compartments, as previously optimized [9]. The CSF level of each chemokine/cytokine was normalized to the corresponding protein content. 

Molecules that were not detected in at least 50% of CSF and serum samples of the entire cohort (MS, OND, HS) were excluded from the analysis [25]: for these reasons, the final panel included 61 molecules. The excluded molecules were LIGHT/TNFSF14, IL-12(p70), IL-19, IL-22, IL-26, IL-27(p28), IL-34, IL-2, CXCL6, CCL27, CCL17, and MMP3. Following a comparison of the two subgroups of OND cases (n = 16 patients with an inflammatory neurological disease and n = 14 patients with a non-inflammatory neurological disease), no statistically significant difference was found in the serum levels of the examined molecules; therefore, the two groups were considered as a single OND group.

### 4.4. Statistical Analysis

The R software was used to perform statistical analysis. A *p*-value < 0.05 was considered statistically significant. Correlations between serum and CSF expression levels for each molecule were evaluated by using the nonparametric Spearman rank correlation. A pairwise univariate Spearman rank correlation index was also used to measure the association between clinical and MRI variables at diagnosis and both serum and CSF protein levels. A false discovery rate (FDR) correction method with a significance level of 0.05 was adopted to correct the multiple-testing problem. In addition, the results from all the multivariate analyses have been adjusted either for the age at the lumbar puncture or for sex.

Differences between serum expression levels among MS patients stratified according to the clinical form at the time of diagnosis (RRMS, PPMS, and CIS) were assessed using a nonparametric Kruskal–Wallis test followed by pairwise post hoc comparison using Benjamini and Hocberg (BH) approach. The Kruskal–Wallis test, followed by the BH method, was applied to compare serum expression levels between MS, OND, and HS.

## Figures and Tables

**Figure 1 ijms-24-03768-f001:**
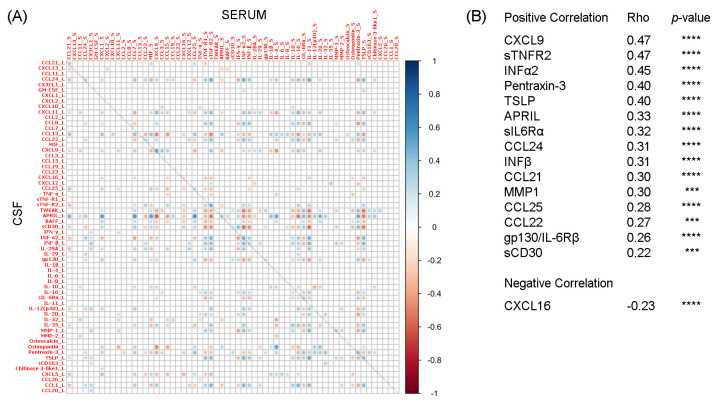
Correlation analysis of liquor and serum expression levels of 61 inflammatory molecules determined by multiplex bead-based immune assays. (**A**) Dots represent Rho-values, and color gradient indicates positive (blue) or negative (red) values. The correlation between each molecule’s liquor and serum expression levels is distributed along the diagonal. (**B**) A panel of 16 molecules whose expression level showed a significant correlation between CSF and serum. Spearman’s correlation followed by FDR correction; *** *p* < 0.001, **** *p* < 0.0001.

**Figure 2 ijms-24-03768-f002:**
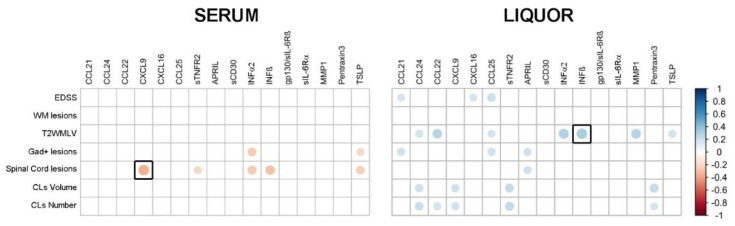
Correlation analysis between clinical parameters at baseline (T0) and the expression levels of the 16 identified molecules in either serum or liquor compartments (Spearman’s method, no correction). Molecules enclosed in the bold squares (CXCL9 for serum and IFNβ for CSF) were the only molecules that correlated significantly with clinical data following FDR correction. WM: white matter; T2WMLV: T2 white matter lesion volume; CL: cortical lesions.

**Figure 3 ijms-24-03768-f003:**
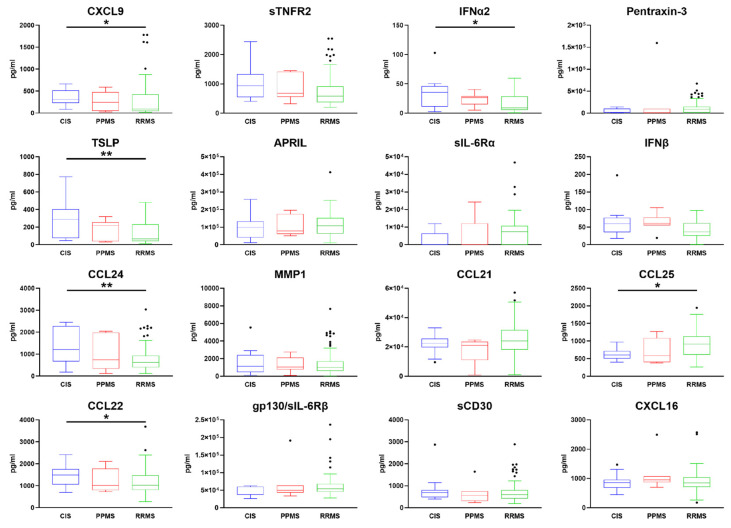
Following stratification of MS patients according to the clinical form at the time of diagnosis (RRMS, n = 123; PPMS, n = 7; CIS, n = 13), the serum expression level of CCL25 was increased in RRMS patients compared to CIS patients. Conversely, CXCL9, IFNα2, TSLP, CCL24, and CCL22, were found to be less expressed in the serum of patients with RRMS compared to CIS. Boxes represent 50% of the central data, shown by a line inside that represents the median. Boxes start in the first quartile (25%) and end in the third (75%). Kruskall–Wallis test followed by Benjamini and Hochberg multiple comparisons test was performed; * *p* < 0.05, ** *p* < 0.01.

**Table 1 ijms-24-03768-t001:** Serum expression levels of the 16 molecules correlated with the CSF expression (see Figure 1). Mean ± SD is reported; for statistical analysis, the Kruskall–Wallis test followed by Benjamini and Hochberg multiple comparisons test was performed, and the adjusted *p*-value is given. Blue and red squares indicate a fold change (FC) >1.3 or <−1.3, respectively; light-blue squares (CXCL16, MS vs. OND and OND vs. HS) indicate an FC = 1.28. Serum expression levels of the remaining 45 inflammatory molecules as well as the CSF expression level of all the analysed molecules are reported in Appendix A. MS: multiple sclerosis; OND: other neurodegenerative diseases; HS: healthy subjects.

Serum Expression Values (pg/mL)	MS	OND	HS	*p*-Value		
	(n = 143)	(n = 30)	(n = 18)	MS vs. OND	MS vs. HS	OND vs. HS
**MIG/CXCL9**	278.7 ± 335.3	166.6 ± 257.7	93.9 ± 76.2	ns	ns	ns
**sTNF-R2**	770.9 ± 517.7	884.4 ± 467.3	440 ± 160.9	ns	0.007	0.0001
**INF-α2**	19.1 ± 16.8	29.7 ± 48	108.8 ± 227.3	ns	0.03	0.02
**Pentraxin-3**	11,681.1 ± 17,389	26,546.9 ± 44,145.5	12,287.8 ± 7533.9	0.0005	ns	ns
**TSLP**	148.2 ± 138.6	31.8 ± 19.7	22.5 ± 18.7	<0.0001	<0.0001	ns
**APRIL/TNFSF13**	111,867.1 ± 65,472.8	135,471.3 ± 116,679.2	174,024 ± 125,614	ns	ns	ns
**sIL-6Rα**	6786.6 ± 7412.3	19,961.4 ± 12,826.3	12,770.6 ± 10,725.5	<0.0001	0.01	0.01
**EOTAXIN-2/CCL24**	829 ± 606.5	671.6 ± 543.2	336.8 ± 310.7	ns	0.0001	0.0001
**INF-β**	46.8 ± 26.4	28.7 ± 14.7	16.7 ± 15	0.0001	0.0001	0.007
**6Ckine/CCL21**	24,651.8 ± 10,681.2	25,802.4 ± 8115.5	20,757.4 ± 13,639.5	ns	ns	ns
**MMP-1**	1419.5 ± 1253.9	3471 ± 9088.8	628.5 ± 816.3	ns	0.005	0.005
**TECK/CCL25**	872.3 ± 344.1	1648.7 ± 840.8	955.3 ± 393	<0.0001	ns	0.003
**MDC/CCL22**	1183.2 ± 528.5	1134.3 ± 652.7	782.7 ± 438.9	ns	0.01	ns
**gp130/sIL-6Rβ**	59,752.2 ± 28,655.5	107,413 ± 59,655.2	69,055.1 ± 48,022.6	<0.0001	ns	0.004
**sCD30/TNFRSF8**	701.7 ± 445.6	1422.4 ± 985.7	1096.7 ± 1452.5	<0.001	ns	0.03
**SCYB16/CXCL16**	908.2 ± 338.8	710.2 ± 261.2	559.1 ± 157.9	0.004	<0.001	0.01

**Table 2 ijms-24-03768-t002:** Clinical data from MS and OND cohorts. MS: multiple sclerosis; OND: other neurological diseases; HS: healthy subjects; CIS: clinically isolated syndrome; RRMS: relapsing-remitting multiple sclerosis; PPMS: primary progressive multiple sclerosis; EDSS: Expanded Disability Status Scale; OCB: the oligoclonal band. The OND group was also subdivided into disorders associated or not with inflammation (Y/N). Mean ± SD is provided for continuous variables (age at PL and Q_alb_); median (range) is provided for EDSS.

	MS (n = 143)	OND (n = 30)
Gender (M/F)	38/105	13/17
Age at PL (years)	39 ± 12.4	46 ± 12.3
MS type (CIS/RRMS/PPMS)	13/123/7	-
EDSS	2 (0–5)	-
OCB (−/+)	44/99	-
Q_alb_ (<7/>7/no data)	111/21/11	-
Inflammatory/Non-inflammatory OtherNeurological Disorders (Y/N)	-	16/14

## Data Availability

Not applicable.

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
