# Peer review of "Intrathecal versus Peripheral Inflammatory Protein Profile in MS Patients at Diagnosis: A Comprehensive Investigation on Serum and CSF"

_ijms, 2023, doi:10.3390/ijms24043768_

Round 1

Reviewer 1 Report

Authors compared 61 inflammatory/immunoregulatory molecules in serum and CSF samples by multiplex immunoassay. There is no detailed clinical characteteristics of patients (the time from last corticosteroids therapy, presence of acute or chronic infections, age and gender, treatment with systemic immunosuppression and so on) which could unspecifically be influenced of the serum markers level. The is no correction of P-values, thus a false positive corrections were not excluded.

Author Response

We thank the Reviewer 1’s comment that helped us to improve the manuscript. The modified parts are written in blue in the revised version. Below we all the responses to the Reviewer 1.

Authors compared 61 inflammatory/immunoregulatory molecules in serum and CSF samples by multiplex immunoassay. There is no detailed clinical characteteristics of patients (the time from last corticosteroids therapy, presence of acute or chronic infections, age and gender, treatment with systemic immunosuppression and so on) which could unspecifically be influenced of the serum markers level.

Reply:

We thank the reviewer for his comments. In the Material and Methods section, we had already specified that i) the 143 patients enrolled in the study were “naïve” (not under treatment at the time of PL) and ii) the collection of samples was performed at least two months after the last relapse, so at least two months after the cortisone treatment. We changed the text in the revised manuscript at line 243 to better explain this second point:

“Paired CSF and serum samples were collected from 143 treatment-naïve patients with MS at the time of diagnosis (and at least two months after the last relapse and the cortisone treatment; Table 2).”

Moreover, following the evaluation of sierological data obtained the day of lumbar puncture (including complete blood cell count, VES and PCR), no patient showed any clinical and humoral signs of ongoing infections.

Following the reviewer’s suggestions, we improved the text at line 244 after citing table 2 and specified the inclusion and exclusion criteria as follows:

“The inclusion criteria to be enrolled in the study were: i) to come to our medical centre consecutively from 2017 to 2020; ii) to receive a confirmed diagnosis of MS. The exclusion criteria were: i) any disease modifying treatment (DMT) or any other therapy targeting the immune system, ongoing at the time of lumbar puncture; ii) concurrence of any immunological or haematological diseases or any chronic infectious; iii) cortisone treatment within two months before the lumbar puncture; iv) pregnancy”.

The is no correction of P-values, thus a false positive corrections were not excluded.

Reply:

We thank the Reviewer 1, we have probably not clearly described the statistical methods that we have now modified and improved. As described in Statistical Analysis section, the correction of p-value has been performed for each analysis (FDR for results in Figure 1 and 2, Benjamini & Hocberg multiple comparisons test for results in Figure 3 and Table 1) and adjusted p-value were reported. We have now better explain this part. In addition, we also outlined that all the results from multivariate analysis have been adjusted for the age at the lumbar puncture and for sex. Accordingly, we add a sentence in the text at line 318: “In addition, the results from all the multivariate analysis have been adjusted either for the age at the lumbar puncture and for sex.”

Reviewer 2 Report

The authors investigated the correlation between CSF and serum levels of 61 inflammatory proteins. However, there are problems that need to be tackled.

1.     OND is a mixed group. The results may largely fluctuate if different diseases were included in this group, except that the authors have a large number of patients. Therefore, I suggest OND can be replaced with NMOSD, MOGAD or some specific disorders.

2.     Please provide the inclusion and exclusion criteria in the method section.

3.     Does the ratio of serum and CSF level of the inflammatory molecule have the predictive value of disease activity or the differential role?

4.     Please list the importance of the probable molecules in the MS patients in predicting the activity or differentiating from others.

5.     Other factors (e.g. age at onset, sex) that may confound the results should be adusted in the analysis.

Author Response

We thank the Reviewer 2’s comment that helped us to improve the manuscript. The modified parts are written in blue in the revised version. Below, all the responses to Reviewer 2:

The authors investigated the correlation between CSF and serum levels of 61 inflammatory proteins. However, there are problems that need to be tackled.

  1. OND is a mixed group. The results may largely fluctuate if different diseases were included in this group, except that the authors have a large number of patients. Therefore, I suggest OND can be replaced with NMOSD, MOGAD or some specific disorders.

Reply:

As suggested we have better detailed some of the most frequent diseases both in the inflammatory and in the not-inflammatory group (line 254). As requested, the inflammatory group mainly consists of NMOSD and MOGAD or patients with ON and/or myelitis. The non-inflammatory group is quite heterogeneous but we believe that this could be considered a strength point of the paper.

  1. Please provide the inclusion and exclusion criteria in the method section.

Reply:

The methods section at line 244 has been updated as requested:

“The inclusion criteria to be enrolled in the study were: i) to come to our medical centre consecutively from 2017 to 2020; ii) to receive a confirmed diagnosis of MS. The exclusion criteria were: i) any disease modifying treatment (DMT) or any other therapy targeting the immune system, ongoing at the time of lumbar puncture; ii) concurrence of any immunological or haematological diseases or any chronic infectious; iii) cortisone treatment within two months before the lumbar puncture; iv) pregnancy”.

  1. Does the ratio of serum and CSF level of the inflammatory molecule have the predictive value of disease activity or the differential role?

Reply:

Following the reviewer suggestion, we calculated the CSF/serum ratio for the 16 molecules showed in Figure 1 and correlated them with the clinical parameters. The obtained results validated those of Figure 2, confirming a positive correlation of IFNα2 and IFNβ with T2WMLV. In addition, we tried to understand whether the CSF/serum ratio of each inflammatory molecule correlated with Qalb, or whether it could discriminate the different clinical phenotypes (CIS, PP, RR), but we did not find any significant results for both analyses.

  1. Please list the importance of the probable molecules in the MS patients in predicting the activity or differentiating from others.

Reply:

Thanks to the Reviewer 2 we have now better described the potential relevant role of the probable molecules detected in the serum to discriminate MS patients with high level of cortical lesions since the time of diagnosis. Considering the high impact of spinal cord damage on disability accumulation this will indeed help, together with imaging tools, to early discriminate MS patients at high risk of rapid progression, especially in MS centres where spinal cord imaging is not available. We have accordingly modified the text of the main manuscript (lines 170 and 207).

  1. Other factors (e.g. age at onset, sex) that may confound the results should be adjusted in the analysis.

Reply:

We thank the reviewer for raising this point. All the presented results from multivariate analysis have been already adjusted for the age at the lumbar puncture and for sex. Nevertheless, we add a sentence in the Statistical Analysis section at line 318 to specify more clearly this point:

“In addition, the results from all the multivariate analysis have been adjusted either for the age at the lumbar puncture and for sex.”

Round 2

Reviewer 1 Report

The absence of clinical characteristics of patients and therapy they are receiving do not allow to exclude bias.

Author Response

We understand the importance of exclude any bias in this study, for this reason we examined paired CSF/serum samples only from a selected MS group including naïve patients at first neurological and radiological assessment, at the time of the clinical onset, and then diagnosed according to the 2010 revision of diagnostic criteria (Polman et al., Ann Neur 2011) without any other inflammatory conditions. Only following the enrolment, the patients started one of the first-line (interferon beta-1a, glatiramer acetate, teriflunomide, and dimethylfumarate) disease-modifying treatments, therefore the CSF/serum protein analysis was not influenced by the treatments. Similarly for the other inflammatory conditions: we collected CSF/serum samples at time of onset, before the diagnosis, in absence of the therapies.

Regarding all the clinical (including the clinical forms) and radiological features of the examined patients, they were evaluated by the neurologists of the group blinded respect to the CSF/serum analyses and when all the data were correlated a false discovery rate (FDR) correction method with a significance level of 0.05 was adopted to correct for the multiple-testing problem and, in addition, the results from all the multivariate analysis have been adjusted either for the age at the lumbar puncture and for sex.

Thanks to the Reviewer 1 suggestion, the part of the MS and OND control population described in the methods was now further modified (in red) in order to improve the manuscript.

Reviewer 2 Report

None.

Author Response

We would like to thank the Reviewer 2 for the useful suggestions.